# Practice and factors associated with pediatrics pain management among nurses working in Bahir Dar city public hospitals: A mixed method study

Bekele Berihun[1], Netsanet Fentahun[2], Lakew Asmare[3], Zeamanuel Anteneh Yigzaw[4]*

1 Department of Nursing, Felege - Hiwot Comprehensive Specialized Hospital, Amhara Regional Health Bureau, Bahir Dar, Ethiopia, 2 Department of Nutrition and Dietetics, School of Public Health, College of Medicine and Health Sciences, Bahir Dar University, Bahir Dar, Ethiopia, 3 Department of Epidemiology and Biostatistics, School of Public Health, College of Medicine and Health Sciences, Wollo University, Dessie, Ethiopia, 4 Department of Health Promotion and Behavioral Science, School of Public Health, College of Medicine and Health Sciences, Bahir Dar University, Bahir Dar, Ethiopia

* zeamanuel19@gmail.com

**Data Availability Statement:** All relevant data are within the manuscript and its supporting information file

## Abstract

### Background

Pain is the most misunderstood, underdiagnosed, and undertreated/untreated medical problem, particularly in children. The main aim of this study was to assess practice and factors associated with pediatrics pain management among nurses working in Bahir Dar city public hospitals, Amhara region, North West Ethiopia, 2022.

### Method

An institutional-based cross-sectional concurrent mixed study design was conducted on randomly selected 421 nurses from November 1 to 30/2022. Purposively selected 8 nurses in different positions and qualifications were included in a qualitative study. A structured self-administered questionnaire and a semi-structured in-depth interview questionnaire were used for data collection. Epi info version 7.1 was used for data entry and SPSS version 25 was used for analysis. ATLAS ti version 7.0 and thematic analysis were used for qualitative study. Binary logistic regression was done to identify predictor variables associated with outcome variables at p <0.05 with a 95% confidence interval. Hosmer and Lemeshow's tests were checked for model goodness of fit, which was 0.71.

### Result

The good practice of pediatric pain management among nurses for hospitalized children was 216 (53.6%) (95% CI- 48.4% to 58.3%). Knowledge [AOR = 3.95; 95%CI: (2.30, 6.79)], attitude [AOR = 2.57; 95% CI: (1.53–4.30)], qualified in BSC pediatrics and child health nurses [AOR = 6.53; 95%CI: (1.56–27.25)], year of experience in pediatrics unit [(AOR = 1.92; 95% CI: (1.03–3.56)] and gating pain management training [AOR = 3.31; 95% CI: (1.73–6.33)] were significant factors. Four themes inadequate knowledge of pain assessment and

**Funding:** The author(s) received no specific funding for this work.

**Competing interests:** The authors have declared that no competing interests exist.

**Abbreviations:** AAPH, Addis Alem Primary Hospital; BDU, Bahir Dar University; BSC, Bachelor of Science; CMHS, College of Medicine and Health Sciences; ETAT, Emergency Triage Assessment and Treatment; FHCSH, Felege Hiwot Comprehensive Specialized Hospital; FLACC, Face, Leg, Activity, Cry and Consolability; ICU, Intensive Care Unit; KM, Kilo Meter; NICU, Neonatal Intensive Care Unit; NSAID, Non-Steroidal Anti Inflammatory Drugs; PCHN, Pediatrics and Child Health Nurse; PRN, Pro Rental Necessary; SPSS, Statistical Package for Social Science; TGCSH, Tibebe Ghion Comprehensive Specialized Hospital; WHO, World Health organization.

management practice, inadequate professional commitment, organization-related factors, and impacts of family knowledge, culture, and economic status were explored.

## Conclusion

Only half of the participants had good practice. Knowledge, attitude, nurses qualified in BSC pediatrics and child health, years of experience in the pediatrics department, and pain management training were associated factors. From the qualitative findings, the unavailability of anti-pain drugs, lack of training, assessment tools, continuous monitoring and evaluation, updated protocols, shortage of resources, and others were the barriers to proper pain management. This study concludes that applying effective pain management practices to hospitalized children remains a challenge. Therefore, it is better to put further effort towards improving pediatric pain management practice.

## Introduction

A revised definition of pain is a distressing experience related to actual or potential tissue damage with sensory, emotional, cognitive, and social components [1]. In line with the definition of the International Association for the Study of Pain, "Pain is an unpleasant sensory and emotional experience related to actual or potential tissue damage" or pain is usually a private experience that's influenced to varying degrees by biological, psychological, and social factors [2].

Pain is the most misunderstood, underdiagnosed, and under-treated/untreated medical problems, particularly in children. One of the foremost challenging roles of medical providers serving children is to appropriately assess and treat their pain. Currently, pain is "the fifth vital sign" and requires caregivers and health professionals to assess regularly to deal with pain. Pain could be a personal experience; many terms are used to describe different sensations. Pain assessment in children is linked to their level of development. Children of identical age vary widely in their perception and tolerance of pain [3]. In 2014, the Federal Ministry of Health (FMOH) with the American Cancer Society launched the Pain-Free Hospital Initiative to integrate pain treatment into service delivery by providing education for hospital staff, raising motivation and awareness, measuring and documenting pain levels, and improving medicine supply [4].

A prospective cross-sectional survey conducted in US hospitals showed that 76% of hospitalized patients had experienced pain during the previous 24 hours, usually acute or procedural pain, and 12% suffered from chronic pain. From those, 20% of children experienced moderate and 30% had severe pain in that period. The worst pain reported by patients was caused by needle pokes (40%), followed by trauma/injury 34% [5].

In Ethiopia, the overall prevalence of pain among hospitalized children was 76.2%, 70% of the children had mild to moderate pain and 6.2% had severe pain based on the age-appropriate assessment tools [6].

Nurses managing their pain ineffectively affects the patient's quality of life negatively and results in higher hospital readmission rates, more repeated outpatient visits, prolonged hospital stays, increased risk of hospital-acquired infection, and increased stress and anxiety for the patient as well as his family. Untreated persistent pain causes several potential complications like anxiety, depression, insomnia, fatigue, poor concentration and short-term memory, stress-related health problems, lack of daily structure and feeling aimless, and also lack of

meaning or direction in life. Similarly, untreated pain leads to biological, psychological, social, developmental, and behavioral problems in children [18].

The factors associated with a low level of practice are; availability of drugs and equipments, availability of protocols and guidelines, work overload, lack of in-service training, lack of pain assessment tools, short duration of experience in pain management, and low level of knowledge and attitude in pediatrics pain management [7–9].

A study conducted at Jimma hospital showed that the overall level of nurses' pain management practice was 36.6% [10]. A study conducted in Black Lion hospital showed that 52.7% of respondents had good practice regarding pediatric pain management [9]. The overall non-pharmacology pain management practice level of nurses in Debre Tabor comprehensive specialized hospital showed that only 44(26%) of nurses had good practice on non-pharmacology pain management methods [11]. A study conducted in the Amhara region referral hospital showed that the practice of nurses; out of a total of 289 respondents, 54.3% had poor practice and 45.7% had good practice [8]. A cross-sectional institution-based study conducted in Bahir Dar City HCPS showed that 76.8% of respondents used the WHO pain management ladder to manage pain. 95.5% of the respondents used paracetamol, and 84.5% of the respondents used NSAIDS to manage pain. The majority of the participants (79.3%) used analgesics during bone marrow aspiration and (72.3%) during wound care [7]. In Ethiopia, little is known about pediatrics pain management practice among nurses in hospitalized children. As many studies showed, pain is unrecognized and undertreated by healthcare professionals all over the world. Thus, managing pain properly is the only option that can prevent children from unnecessarily staying in hospitals and suffering from pain. In Ethiopia as well as in the Amhara regional state, little is known about the practice and factors associated with pediatric pain management among nurses. Therefore, the result of this study was used to highlight the gaps in pain management practice and help to identify factors affecting pain management practice.

Therefore, this research aimed to assess the practice and factors associated with pediatric pain management for hospitalized children among nurses working in Bahir Dar city public hospitals using a mixed-study design.

## Methods and materials

### Study design

Institutional-based cross-sectional concurrent mixed study design was used.

### Study area and period

The study was conducted at Bahir Dar city public hospitals, Amhara regional state, North West Ethiopia. Currently, from the three public hospitals in the city 834 [474 at FHCSH, 310 at TGCSH, and 50 at AAPH] nurses were found in different qualifications. Of those nurses, 733[380 at FHCSH, 303 at TGCSH, and 50 at AAPH] were working at in-patient departments and those have contact with a variety of patients with pain. The study was conducted between November 1 to 30/ 2022.

### Sample size determination

For the quantitative part: The sample size was determined by using a single population proportion formula, taking the proportion as 45.7% from a previous study conducted in public referral hospitals of Amhara region, Ethiopia (8), 95% confidence interval (CI) and 5% margin of error. Thus, the sample size is calculated as follows: -

Hence, $n = \frac{(z\,a/2)^2 * P(1-p)}{(d)^2}$

Where: **n** = required sample size

**Zα/2** = critical value for normal distribution at 95% confidence interval which equals 1.96

**P** = 45.7% (The proportion of nurses who had good practice on pain management taken from a previous related study conducted at Amhara region public referral hospitals [8].

**d** = margin of error 5%

$$\mathbf{n} = \frac{(1.96)^2 0.457(1 - 0.457)}{(0.05)^2}$$

**n = 382** (by adding 10% non-response rate the sample size was)

**n = 421** (final sample size)

Whereas, for the qualitative part: The sample size was determined by saturation of the data that needs to be explored. A person was asked to participate in interviews until additional interviews did not provide additional evidence about the main themes of interest.

## Sampling technique

In this study, a simple random sampling technique was used. The list of nurses was obtained from human resource administrators and to know nurses working in the in-patient department from each hospital, the list was obtained from the nurse's director (metron) then the study participant was selected by using the lottery method until fulfilled sample size (**Fig 1**).

For the qualitative part: a heterogeneous purposive sampling technique was used to recruit in-depth interview participants. The in-depth interview included different qualifications of nurses.

## Populations

**Source population.**   All nurses working in Bahir Dar city public hospitals in in-patient departments.

**Study population.**   For the quantitative: Sample nurses who work at in-patient departments and are available during the data collection period.

For qualitative: purposively selected nurses working in in-patient departments.

## Inclusion and exclusion criteria

**Inclusion criteria.**   Nurses who work in in-patient departments in Bahir Dar city public hospitals during the data collection period and had more than 6 months of work experience was included in the study.

**Exclusion criteria.**   Individuals who were seriously ill during the data collection period were excluded from the study.

## Study variables

### Dependent variables

- Practice of pediatrics pain management

### Independent variables

- **Demographic characteristics:** - Age, sex, level of education, working experience and working unit.

- **Nurses-related factors**: knowledge and attitude

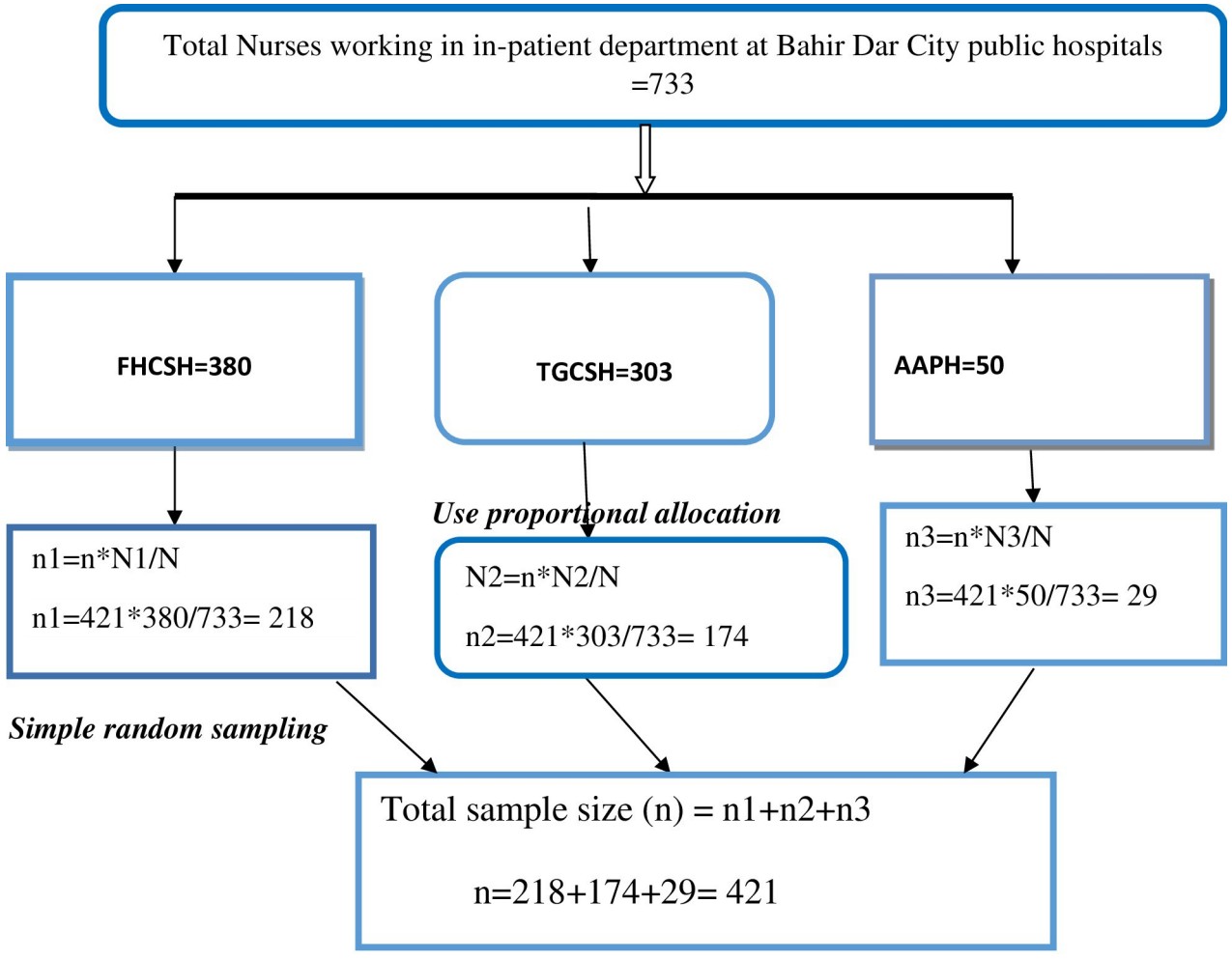

**Fig 1. Schematic presentation of sampling technique to assess pediatrics pain management practice and factors associated with hospitalized children among nurses working in Bahir Dar city public hospitals, Northwest Ethiopia, mixed method, 2022.**

- **Patient-related factors**: Ability to communicate, level of consciousness, patient/family need for drugs, age, culture, language, and lack of awareness

- **Organizational factors:** pain management protocols of the hospital, pain management guidelines, availability of drugs, work overload, in-service training, and equipment/materials.

## Operational definitions

**Pediatrics:** Neonates, infants, and adolescents are less or equal to 18 years [7].

 **Pain management**: the alleviation of pain or a reduction in pain to a level that is acceptable to the client by using pharmacologic and non-pharmacologic methods.

 **Good practice** - nurses who scored above the mean (11.36) had good practice in pain management [8].

 **Poor practice**–those who scored below the mean (11.36) had poor practice in pain management.

**Good knowledge** - those nurses who scored above the mean (19.78) had good knowledge of pain management [8].

**Poor knowledge**- those who scored below the mean (19.78) for knowledge items.

**Favorable attitude**- those nurses who scored above the mean (10.79) had a favorable attitude [8].

**Unfavorable attitude** -nurses who scored below the mean (10.79) on attitude items.

**Non-pharmacological pain management**: therapies that help to decrease pain without the use of medications.

**Hospitalized children:** children admitted to the governmental institution to get medical attention from HCPs [12].

## Data collection tool and procedure

**Data collection tool.** The questionnaire was adapted by using different literatures related to this topic. For quantitative data: a structured self-administered English version questionnaire was used. It has five parts; the first section is socioeconomic variables (08 items). The second section contains questions that can measure the practice of nurses for pain management (19 items). The other section contains questions that can measure the knowledge of nurses for pain management (14 major items), questions that can measure the attitude of nurses toward pain management (15 items), and questions that can measure organization and patient-related factors for pain management in hospitalized children (12 items).

For qualitative data: semi structured in-depth interview was used. The principal investigator was the data collector. The interview was initiated with general questions followed by probing questions, and then the questions became more focused as the data collection progressed.

**Data collection procedure.** Data collection was facilitated by seven bachelor of science (BSC) holder nurses (six data collectors and one supervisor) after taking two-day training to familiarize the data collection procedure and instrument on the objective and relevance of the study. Data collectors were supervised at each site. The principal investigator and the supervisors collected the filled questionnaire and checked for missed values and completeness daily.

For qualitative: semi-structured, face-to-face in-depth interviews having open-ended questions used a guide, tape recorder, and notebooks were employed among nurses working at public hospitals. The modulator and the principal investigator were introduced to the purpose of the study, assured them about confidentiality, and took informed verbal consent. The study setting was chosen such that the privacy of the participants could be maintained, the location was comfortable, noisy areas were avoided, and the location should be easily accessible.

## Data quality control

Two-day training was given for both data collectors and supervisors. Before the actual data collection, the items were tested on 21 (5% of 421) nurses working out of the study area (Debre Tabor hospital) and the result was used to check clarity and completeness, understandability; flow and construction, internal consistency of questionnaire and those questions found to be unclear or confusing have been modified based on the result of the pretest.

The principal investigator and trained supervisor supervised and reviewed every data collection procedure, and questionnaire for completeness and logical consistency and correction made. The principal investigator collected the completed questionnaires every day and was responsible for the coordination and on-the-spot supervision of the overall data collection process. The principal investigator was performing data coding, entry, and cleaning.

## Data processing and analysis

First data was checked for completeness and then a unique code was assigned for each completed questionnaire. Subsequently, data entered using Epi-info version 7.1. The generated data was exported to a statistical package for social sciences (SPSS) version 25. Analysis was done with descriptive statistics by using frequency, percentage, and mean. Bivariate and multivariate analysis between dependent and independent variables was performed using logistic regression. Variables with a p-value less than 0.25 in the bivariable analysis were taken as the candidates for the multivariable logistic regression model. Logistic regression model fitness was checked using Hosmer & Lemeshow's goodness of fit test, which was 0.71. During the analysis, crude and adjusted odds ratios at 95% confidence interval (CI) and p-value less than 0.05 were considered. A p-value of less than 0.05 was taken as a significant association; results were present in text, tables, charts, and graphs.

Quantitative data is triangulated with qualitative data. The qualitative data was analyzed through the thematic analysis using Atlas ti version 7.0 software. Major and subthemes were created from the text itself through repeated reading. After reading the transcripts, the investigator identified emergent themes and then coded each theme to delineate individual topics. Statements are grouped by code to the corresponding theme. The findings were present in narratives by thematic areas and support the quantitative data.

## Trustworthiness and rigor of the study

For the qualitative part, the guides were originally developed in English and then translated into the local language Amharic for data collection. After data collection, verbatim transcription was applied to transcribe the audio-record interview into Amharic and then again translated into English for analysis. When necessary, the transcripts were supplemented by field notes to clarify issues. In the meantime, English transcripts were read many times to develop codes that identify important and common concepts that relate to the main themes of the study. The principal investigators used Atlas ti software for coding. Each transcript was screened, quoted, and coded. The codes in turn grouped into sub-themes and then into main themes.

## Credibility

To maintain credibility, before the data collection, the investigators evaluated the in-depth interview questions. All members who participated in the study took orientation about the purpose of the study and the responsibility given before the interview took place to avoid unnecessary interruption and keep the rights of the participants. Professionals before data collection were evaluated the interview guide.

## Transferability

The transferability of the study was achieved by describing the study setting, sample, and data collection procedure clearly and in detail. The interview was conducted with the aid of digital audio records and field notes for observational data.

## Dependability

The dependability of the study was obtained through prolonged engagement with the participants to develop trust in the interviewer. This helps to increase the consistency of the study.

## Conformability

The conformability of the study was ensured by a detailed recording of each activity of the participant at the time of the interview and every procedure of the study. The principal investigator participated from the start to the end of the study.

## Ethics approval and consent to participate

Ethical approval and clearance for this study were obtained from the Institutional Review Board (IRB) of Bahir Dar University, College of Medicine and Health Science, and School of Public Health on November 1/ 2022 with Ref. No /5125/2.4. At all levels, letters of cooperation were given to the relevant administrative officials. Informed consent was obtained from the study participants.

## Results

### Quantitative finding

**Socio-demographic characteristics of the study participants.**    A total of 421 nurses participated with a 95.7% response rate. The analysis showed that 261(64.8%) were females. The age of the study respondents ranged from 22 to 55 years with a mean age of 32 ± 5.1 years. Almost half of the participants 197(48.9%) were in the age group of 31–40 years. From the participants 162(40.2%) had 5–10 years and 113(28.0%) had more than 10years' experience. Similarly, of the participants, 138(34.2%), 160(39.7%), and 76(18.9%) were less than two years, 2–5 years, and 5–10 years worked at pediatric ward and ETAT respectively. Out of the total participants, 336 (83.3%) were qualified for degree levels in different nursing professions whereas, 33 (7.7%) were diploma nurses, while the rest 36(9.0%) were masters and others. Among those more than half of the total participants 266 (66.0%) were BSC in comprehensive nursing and 70(17.3%) were BSC in PCHN and neonatal nursing (**Table 1**).

Ninety-one (22.6%), 84(20.8%), 64(15.9%), 39(9.7%), 26(6.5%), and 31(7.7%) participants were currently working in the pediatric ward, emergency/ETAT, NICU/ICU, surgical ward, orthopedic ward, and medical ward unit respectively. While 68(16.9%) were from other inpatient departments of the hospitals (**Fig 2**).

**Nurses' knowledge about pediatric pain management.**    The score of respondents' knowledge of pediatrics pain management practice was added up and dichotomized into two, based on the mean knowledge score, which was above 19.78 with (SD: ±4.10789) were considered as having good knowledge regarding pediatrics pain management practice. The minimum and maximum scores were 5 and 27 respectively. Based on this cut-off point, 233(57.8%) (95%CI 52.6% to 62.8%) of the study participants had good knowledge whereas, 170(42.2%) (95% CI 37.2% to 47.4%) were poor knowledge regarding pediatric pain management practice for hospitalized children (Table 2).

**Attitude of nurses regarding pediatrics pain management.**    In this study, more than half of 225(55.8%) (95%CI 51.1% to 61.2%) of the participants were favorable attitudes whereas 178(44.2%) (95%CI 38.8% to 48.9%) were unfavorable attitudes towards pediatrics pain management for hospitalized children's (**Table 3**).

**Organizational factors towards pediatrics pain management.**    The majority of participants 315(78.2%) did not gate in service training about pediatric pain management. Similarly, 305(75.7%) of participants said that lack of training affects the ability to manage pain effectively. More than half of the participants 263(65.3%) said that no designated area and material affects the ability to manage pain using pharmacological and non-pharmacological methods (**Table 4**).

**Table 1. Socio-demographic characteristics of pediatrics pain management practice and factors associated with hospitalized children among nurses working in Bahir Dar city public hospitals, Northwest Ethiopia, Mixed Method, 2022 (N = 403).**

| Variables | Response | Frequency (N = 403) | Percentage (%) |
|---|---|---|---|
| Age | 20–30 | 185 | 45.9% |
| | 31–40 | 197 | 48.9% |
| | 41–50 | 18 | 4.5% |
| | >50 | 03 | 0.7% |
| Sex | Male | 142 | 35.2% |
| | Female | 261 | 64.8% |
| Years of experience in nursing | <2 years | 33 | 8.2% |
| | 2–5 years | 95 | 23.6% |
| | 5–10 years | 162 | 40.2% |
| | >10 years | 113 | 28.0% |
| Educational status | Diploma | 31 | 7.7% |
| | BSC comprehensive Nurse | 266 | 66.0% |
| | BSC in PCHN | 40 | 9.9% |
| | BSC in Neonatal Nurse | 30 | 7.4% |
| | MSC and above | 22 | 5.5% |
| | Others (Emergency nurses, OR nurses) | 14 | 3.5% |
| Years of experience in the pediatrics unit | < 2 years | 138 | 34.2% |
| | 2–5 years | 160 | 39.7% |
| | 5–10 years | 76 | 18.9% |
| | >10 years | 29 | 7.2% |
| Monthly income | 3201–5800 birr | 32 | 7.9% |
| | 5801–7800 birr | 204 | 50.6% |
| | 7801–10900 birr | 165 | 40.9% |
| | >10900 birr | 2 | 0.5% |

**Patient-related factors towards pediatrics pain management.** The majority of the participants 274(68.0%), 278(69.0%), 292(72.5%), 245(60.8%), and 271(67.2%) reported that patient's inability to communicate, language differences, parents present during pain management, cultural belief and awareness of the family or Childs were an impact on pain management practice respectively (**Table 5**).

**The practice of nurses in pediatrics pain management.** Of the 403 respondents, 326 (80.9%) reported that they assess pain for children, but only 138(34.2%) use a pain assessment tool, from those 75 (18.6%) participants use the tool routinely. Of the participants, only 114 (28.3%) use the self-reported pain scale (FACE scale), while 91(22.6%) use the behavioral pain scale (FLACC scale). Two hundred twenty (54.6%) of nurses are who administer pain medication by their adjustment and 291 (72.2%) of nurses administer pain medication to relieve pain when needed (PRN) for children. Of the participants 269 (66.7%) nurses documented the findings after pain assessment and management while the rest were not documented. The majority of the participants should not assess and administer analgesia before repositioning 368 (91.3%) and endo-tracheal suctioning 280(69.5%). Whereas, 199 (49.4%), and 277(68.7%) nurses were assessed and administered anti-pain drugs during pre and postoperative care and wound care respectively (**Table 6**).

This study shows that, 257(63.8%) participants reported that they use non-pharmacological pain management from those 198 (49.1%) use hot/cold compress, 76(18.9%) use music, 91 (22.6%) use playroom to forget the child's pain, 47 (11.7%) apply pressure and only 6 (1.5%) advice the family to prayers (**Fig 3**).

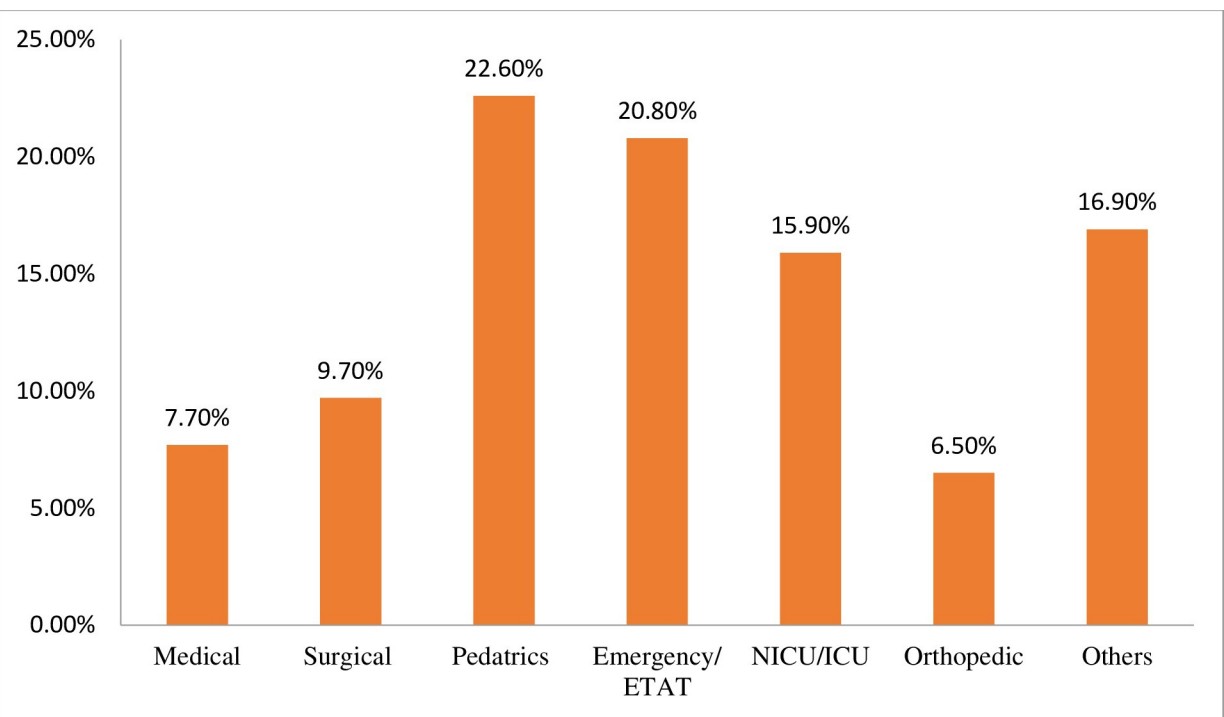

**Fig 2. Percentage distributions of participants by current working unit of pediatrics pain management practice and factors associated with hospitalized children among nurses working in Bahir Dar city public hospitals, Northwest Ethiopia, mixed method, 2022.**

**Table 2. Knowledge of nurses regarding pediatrics pain management practice working in Bahir Dar city public hospitals of Amhara regional state, Ethiopia, 2022 (N = 403).**

| Variables | Response | Frequency (N = 403) | Percentage (%) |
|---|---|---|---|
| Drugs used to treat moderate and severe pain in the pediatric age group. | Incorrect | 125 | 31.0% |
| | Correct | 278 | 69.0% |
| The recommended route of administration of opoid analgesics to children with brief, severe pain of sudden onset | Incorrect | 141 | 35.0% |
| | Correct | 262 | 65.0% |
| A lack of pain expression does not mean a lack of pain | Incorrect | 227 | 56.3% |
| | Correct | 176 | 47.7% |
| Distraction, for example, by the use of music or relaxation, can decrease the feeling of pain | No | 61 | 15.1% |
| | Yes | 342 | 84.9% |
| Increasing analgesic requirements are signs that the patient is becoming addicted to the narcotic | No | 69 | 17.1% |
| | Yes | 334 | 82.9% |
| Patients having severe chronic pain often need higher dosages of pain medications than patients with acute pain | No | 89 | 22.1% |
| | Yes | 314 | 77.9% |
| If a patient (and/or family member) reports that a narcotic is causing euphoria, she should be given a lower dose of the analgesic | No | 116 | 28.8% |
| | Yes | 287 | 71.2% |
| Do you know the consequences of unrelieved/ untreated pain | No | 72 | 17.9% |
| | Yes | 331 | 82.1% |
| Narcotics can cause respiratory depression; therefore, they should not be used in pediatric patients | Incorrect | 218 | 54.1% |
| | Correct | 185 | 45.9% |
| Giving narcotics on a regular schedule is preferred over a PRN schedule for continuous pain | No | 103 | 25.6% |
| | Yes | 300 | 74.4% |

**Table 3. Attitude of nurses regarding pediatrics pain management practice working in Bahir Dar city public hospitals of Amhara regional state, Ethiopia, 2022 (N = 403).**

| Variable | Agree N (%) | Not sure N (%) | Disagree N (%) |
|---|---|---|---|
| Infants and children experience pain equal to that experienced by adults. | 161 (40.0%) | 50 (12.4%) | 192 (47.6%) |
| Children need better attention than adults to manage their pains. | 276 (68.5%) | 39 (9.7%) | 88 (21.8%) |
| Parents should be present during painful procedures. | 191 (47.4%) | 55 (13.6%) | 157 (39.0%) |
| Pain management and pain relief are priorities in children's treatment. | 269 (66.7%) | 26 (6.5%) | 108 (26.8%) |
| Children have the right to appropriate assessment and management of their pain. | 245 (60.8%) | 34 (8.4%) | 124 (30.8%) |
| The most accurate judge of the intensity of the children's pain is patients. | 177 (43.9%) | 70 (17.4%) | 156 (38.3%) |
| For a better assessment of a child's pain, the nurse can discuss it with her/his parents. | 248 (61.5%) | 41 10.2%) | 114 28.3 |
| Assessment and control of child pain led to improve his/her parent's satisfaction. | 221 (54.8%) | 51 (12.7%) | 131 (32.5%) |
| Failure to assess and manage the child's pain affects his body and mind in the long term. | 192 (47.6%) | 55 (13.6%) | 156 (38.7%) |
| The nurse's physical and mental fatigue can affect children's pain relief. | 200 (49.6%) | 62 (15.4%) | 141 (35.0%) |
| Ensuring patient comfort and pain relief is one of the most important tasks of nurses. | 231 (57.3%) | 46 (11.4%) | 126 (31.3%) |
| Communicating with and educating a child's parent plays an effective role in relieving pain. | 225 (55.8%) | 54 (13.4%) | 124 (30.8%) |
| Nurses can provide the most accurate rating of pain intensity and manage pain. | 222 (55.1%) | 41 (10.2%) | 140 (34.7%) |
| Evaluation and measurement of a child's pain should be considered as one of the vital signs when examining the child | 246 (61.0) | 40 (9.9%) | 117 (29.0%) |
| Measurement and control of pain in child leads to improved quality of child's life | 227 (56.3) | 31 (7.7%) | 145 (36.0%) |

The findings of this study also showed that 143(35.5%) and 177 (43.9%) of the participants reported that pain score and management findings are not discussed during nurse-to-nurse reports and unit rounds as a whole. Of the participants, 171(42.4%) reported that they did not agree with children's statements about their pain.

In this study, the overall practice level of nurses was, out of 403 participants 216(53.6%) had good practice whereas, 187(46.4%) participants had poor practice with a mean score of 11.36 ±4.9 (**Fig 4**).

**Factors affecting pediatrics pain management practice.** For the inferential analysis of the data, bi-variable and multivariable logistic regressions were done by using binary logistic regression.

According to bi-variable analysis, years of experience in nursing, qualification, year of experience in pediatrics unit, good knowledge and favorable attitude of nurses, availability of anti-pain drugs, nurses workload, availability of pain management tools, gating pain management training, having specific pain management protocol in the institutions, no designated area and material for pain management, problems due to awareness of families as well as the child and children cooperate in managing pain having history on pain management were eligible for multivariable analysis (p<0.25). Subsequently, these variables were entered into the multivariable analysis

**Table 4. Descriptive frequency of organizational factors of nurses worked in Bahir Dar city public hospitals of Amhara regional state, Ethiopia, 2022 (N = 403).**

| Variables | Response | Frequency (N = 403) | Percentage (%) |
|---|---|---|---|
| Having specific protocols for pediatric pain management | No | 207 | 51.4% |
| | Yes | 196 | 48.6% |
| Do you gate in-service training on pain management? | No | 315 | 78.2% |
| | Yes | 88 | 21.8% |
| Lack of availability of anti-pain drug | No | 74 | 18.4% |
| | Yes | 329 | 81.6% |
| Nursing workload | No | 106 | 26.3% |
| | Yes | 297 | 73.7% |
| Lack of availability of pain assessment tools | No | 142 | 35.2% |
| | Yes | 261 | 64.8% |
| Lack of training affects your ability to manage pain | No | 98 | 24.3% |
| | Yes | 305 | 75.7% |
| Lack of protocols for pain management | No | 145 | 36.0% |
| | Yes | 258 | 64.0% |
| Low priority of pain management by the unit team | No | 192 | 47.6% |
| | Yes | 211 | 52.4% |
| No designated area or material for managing pain | No | 140 | 34.7% |
| | Yes | 263 | 65.3% |

In multivariable binary logistic regression analysis only the knowledge and attitude of participants, qualified in BSC pediatrics and child health nurses, year of experience in the pediatrics department, and gating pain management training were significantly associated with the pain management practice of nurses (p<0.05).

Based on findings from the multivariable binary logistic regression, in this study nurses who had good knowledge were four times [AOR = 3.95; 95% CI: (2.30, 6.79)] more likely to

**Table 5. Descriptive frequency patient-related factors of nurses worked in Bahir Dar city public hospitals of Amhara regional state, Ethiopia, 2022 (N = 403).**

| Variables | Response | Frequency(N = 403) | Percentage (%) |
|---|---|---|---|
| Do you think the Patient's inability to communicate has an impact on pain management? | No | 129 | 32.0% |
| | Yes | 274 | 68.0% |
| Do you think children's consciousness level is decrease and has an impact on pain management? | No | 140 | 34.7% |
| | Yes | 263 | 65.3% |
| Do you think being a child (age) has an impact on the management of pain? | No | 130 | 32.3% |
| | Yes | 273 | 67.7% |
| Do you think language differences have an impact on the management of pain? | No | 125 | 31.0% |
| | Yes | 278 | 69.0% |
| Do you think parents affect your ability to manage the pain of children properly during pain management practice? | No | 111 | 27.5% |
| | Yes | 292 | 72.5% |
| Do you face problems during pain management of the child due to family/ patients' needs for drugs wrongly? | No | 157 | 39.0% |
| | Yes | 246 | 61.0% |
| Do you face problems during pain management for the child due to cultural beliefs? | No | 158 | 39.2% |
| | Yes | 245 | 60.8% |
| Do you face problems during pain management of the child due to awareness of families or children? | No | 132 | 32.8% |
| | Yes | 271 | 67.2% |
| Do children cooperate in managing pain and have a history of pain management? | No | 194 | 48.1% |
| | Yes | 209 | 51.9% |

**Table 6. Descriptive frequency by their assessment and practice level for nurses worked in Bahir Dar city public hospitals of Amhara regional state, Ethiopia, 2022 (N = 403).**

| Variables | | Response | Frequency (n = 403) | Percentage (%) |
|---|---|---|---|---|
| Do you assess pain in children? | | Yes | 326 | 80.9% |
| | | No | 77 | 19.1% |
| Do you use a self-reported pain scale (FACE scale) for the assessment? | | Yes | 114 | 28.3% |
| | | No | 289 | 71.7% |
| Do you use a behavioral pain scale (FLACC) for assessment? | | Yes | 91 | 22.6% |
| | | No | 312 | 77.4% |
| Do you administer pain medication to children by your adjustment? | | Yes | 220 | 54.6% |
| | | No | 183 | 45.4% |
| Do you administer additional pain medication to relieve pain when PRN? | | Yes | 291 | 72.2% |
| | | No | 112 | 27.8% |
| Do you reassess children's pain after giving pain medication? | | Yes | 265 | 65.8 |
| | | No | 138 | 34.2% |
| Do you give post-operative analgesics around the clock on a fixed schedule? | | Yes | 260 | 64.5% |
| | | No | 143 | 35.5% |
| Do you give children sterile water by injection (placebo) to determine if the pain is real? | | Yes | 21 | 5.2% |
| | | No | 382 | 94.8% |
| Do you adjust subsequent doses in accordance? with the individual /patient's response? | | Yes | 218 | 54.1% |
| | | No | 185 | 45.9% |
| Do you check and report to physicians if there are side effects during administering Opioid analgesics? | | Yes | 264 | 65.5% |
| | | No | 139 | 34.5% |
| Do you document the findings after pain assessment and management for patients? | | Yes | 269 | 66.7% |
| | | No | 134 | 33.3% |
| Do you discuss pain scores and management" during nurse-to-nurse reports? | | Yes | 260 | 64.5% |
| | | No | 143 | 35.5% |
| Are pain scores and management discussed during unit rounds? | | Yes | 226 | 56.1% |
| | | No | 177 | 43.9% |
| Do you always agree with children's statements about their pain? | | Yes | 232 | 57.6% |
| | | No | 171 | 42.4% |
| Do you assess and administer analgesia during the following procedures are doing | wound care | Yes | 277 | 68.7% |
| | | No | 126 | 31.3% |
| | endo-tracheal suctioning | Yes | 123 | 30.5% |
| | | No | 280 | 69.5% |
| | pre- and post-operatively | Yes | 199 | 49.4% |
| | | No | 204 | 50.6% |
| | patient repositioning | Yes | 35 | 8.75% |
| | | No | 368 | 91.3% |

practice pediatrics pain management compared to those who had poor knowledge. Similarly, nurses who had favorable attitudes toward pain management were three times [AOR = 2.57; 95% CI: (1.53, 4.30)] more likely able to practice pain management for hospitalized children than those who had unfavorable attitudes.

In addition, BSC pediatrics and child health nurses were seven times [(AOR = 6.53; 95%CI (1.56, 27.25)] more likely to practice pediatrics pain management than nurses who qualified as diploma nurses. Similarly, nurses who had 2–5 years and 5–10 years of experience in pediatrics unit were two times (AOR = 1.92;95%CI:(1.03,3.56), and five times (AOR = 5.41;95%CI:

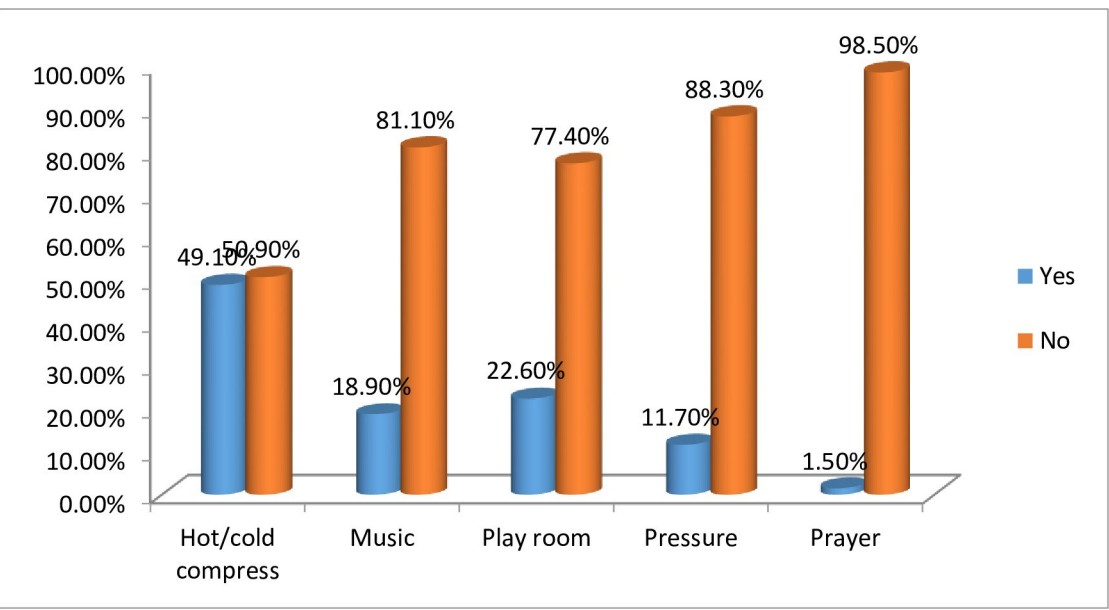

**Fig 3. Percentage distribution of respondents using non-pharmacological pain management of nurses worked in Bahir Dar city public hospitals of Amhara regional state, Ethiopia, 2022 (N = 403).**

(2.42,12.08) more likely able to practice pediatrics pain management compared to those nurses who had less than two years of experiences in pediatrics unit respectively.

Finally, those respondents who received in-service training about pediatrics pain management were three times [AOR = 3.31; 95% CI: (1.73, 6.33) more likely to practice pain management for hospitalized children than those who did have not in-service training on pain management (**Table 7**).

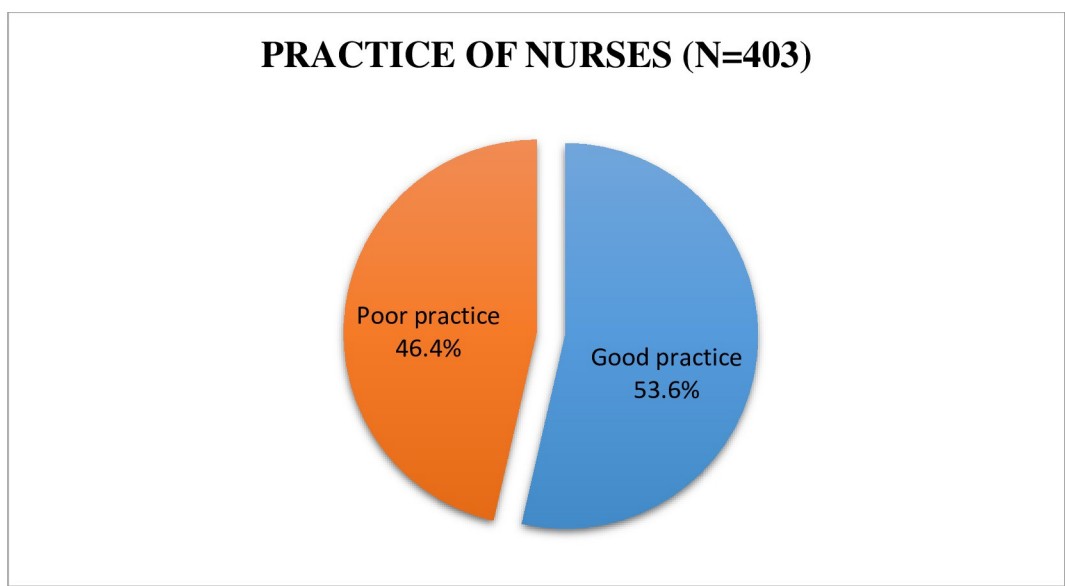

**Fig 4. Total distribution of respondents according to practice of pain management among nurses in Bahir Dar city public hospitals, Amhara region, Ethiopia, 2022.**

**Table 7. Bivariate and multivariable analysis of factors associated with pain management practice in Bahir Dar city public hospitals, Amhara region, Ethiopia, 2022.**

| Variables | Category | Pain management practice | | COR (95% CI) | AOR (95% CI) | P-value |
|---|---|---|---|---|---|---|
| | | Good | Poor | | | |
| Years of experience in nursing | <2 years | 12 | 21 | 1 | 1 | |
| | 2–5 years | 50 | 45 | 1.94(0.86,4.39) | 1.22(0.43,3.40) | 0.703 |
| | 5–10 years | 94 | 68 | 2.41(1.11,5.25) * | 1.43(0.53,3.88) | 0.478 |
| | >10 years | 60 | 53 | 1.98(0.89,4.40) | 1.04(0.35,3.08) | 0.933 |
| Qualification | Diploma | 5 | 26 | 1 | 1 | |
| | BSC comprehensive nurses | 138 | 128 | 5.60(2.09,15.04) * | 2.54(0.84,7.65) | 0.097 |
| | BSC pediatrics nurses | 33 | 7 | 24.51(6.97,86.20) * | 6.53(1.56,27.25) | **0.010**\*\* |
| | BSC neonatal nurses | 19 | 11 | 8.98(2.67,30.16) * | 2.34(0.58,9.36) | 0.229 |
| | Masters and above | 14 | 8 | 9.10(2.49,33.14) | 1.39(0.30,6.49) | 0.669 |
| | Others | 7 | 7 | 5.20(1.25,21.49) | 1.72(0.32,8.98) | 0.521 |
| Years of experience in the pediatrics unit | <2 years | 51 | 87 | 1 | 1 | |
| | 2–5 years | 88 | 72 | 2.08(1.30,3.32) * | 1.92(1.03,3.56) | **0.038**\*\* |
| | 5–10 years | 56 | 20 | 4.77(2.57,8.84) * | 5.41(2.42,12.08) | <**0.001**\*\* |
| | >10 years | 21 | 8 | 4.47(1.84,10.84) * | 3.12(1.00,9.74) | 0.50 |
| Knowledge of nurses | Good knowledge | 159 | 74 | 4.26(2.79,6.49) * | 3.95(2.30,6.79) | <**0.001**\*\* |
| | Poor knowledge | 57 | 113 | 1 | 1 | |
| Attitude of nurses | Favorable attitude | 154 | 71 | 4.05(2.67,6.16) * | 2.57(1.53,4.30) | <**0.001**\*\* |
| | Unfavorable attitude | 62 | 116 | 1 | 1 | |
| Having a protocol for pediatric pain management | No | 94 | 113 | 1 | 1 | |
| | Yes | 122 | 74 | 1.98(1.33,2.95) * | 1.57(0.95,2.60) | 0.075 |
| Do you gate in-service training on pain management | No | 148 | 167 | 1 | 1 | |
| | Yes | 68 | 20 | 3.83(2.22,6.61) * | 3.31(1.73,6.33) | <**0.001**\*\* |
| Lack of availability of anti-pain drugs | No | 33 | 41 | 1 | 1 | |
| | Yes | 183 | 146 | 1.55(0.93,2.58) * | 0.65(0.31,1.37) | 0.268 |
| Nursing workload | No | 46 | 60 | 1 | 1 | |
| | Yes | 170 | 127 | 1.74(1.11,2.73) * | 1.50(0.80,2.81) | 0.200 |
| Lack of availability of pain assessment tools | No | 69 | 73 | 1 | 1 | |
| | Yes | 147 | 114 | 1.36(0.90,2.05) * | 0.94((0.54,1.64) | 0.837 |
| No designated area and material for pain management | No | 68 | 72 | 1 | 1 | |
| | Yes | 148 | 115 | 1.36(0.90,2.05) * | 1.17(0.64,2.12) | 0.596 |
| Facing a problem due to awareness of families and child | No | 61 | 71 | 1 | 1 | |
| | Yes | 155 | 116 | 1.55(1.02,2.36) * | 0.93(0.52,1.66) | 0.825 |
| Children cooperate in managing pain and have a history of pain | No | 87 | 107 | 1 | 1 | |
| | Yes | 129 | 80 | 1.98(1.33,2.95) * | 1.42(0.84,2.42) | 0.184 |

COD- crude odds ratio, AOD-adjusted odds ratio, ** Remained statistically significant (p < 0.05) in adjusted odds ratio.

## Qualitative finding

A qualitative approach was applied to explore the barriers affecting pediatric pain management practice for hospitalized children to support the quantitative study and to explore the factors not addressed by the quantitative study.

**Socio-demographic characteristics of the study participants.** The analysis showed that many of the participants were males and all of the participant's ages were above 30 years. Similarly, all participants' experiences in nursing were more than five years (**Table 8**).

**Table 8. Socio-demographic characteristics of nurses who participated in a qualitative study in Bahir Dar city public hospitals of Amhara regional state, Ethiopia, 2022 (N = 8).**

| Code | Age | Sex | Marital status | Educational level | Position /Work unit/ | Work experience in nursing | Work experience in a pediatrics unit |
|------|-----|-----|----------------|-------------------|----------------------|----------------------------|--------------------------------------|
| P1 | 38 | M | Married | BSC Nurse & MPH | Pedi-ward coordinator | 10 years | 6 years |
| P2 | 39 | F | Married | BSC Nurse | Ward Pain focal | 15 years | 6 years |
| P3 | 37 | M | Married | Pediatrics Nurse | ETAT coordinator | 10 years | 3 years |
| P4 | 35 | M | Married | BSC Nurse | ETAT | 8 years | 2 years |
| P5 | 30 | M | Married | BSC Nurse | Ward | 10 years | 2 years |
| P6 | 30 | F | Single | BSC Nurse | ETAT | 8 years | 2 years |
| P7 | 34 | M | Single | BSC Nurse | Ward | 6 years | 3 years |
| P8 | 36 | M | Single | Pediatrics Nurse | ETAT vice /coordinator | 10 years | 5 years |

**Themes.** Four themes and eight sub-themes were identified by using the qualitative study findings. Those themes are inadequate knowledge of pain assessment and pain management practice, inadequate professional commitment and satisfaction; and organization-related factors and knowledge, culture, and economic status of the family-affected Pediatrics Pain Management practice for hospitalized children (**Table 9**).

**Theme 1: Inadequate Knowledge on pain assessment and management practice.** All participants expressed their clinical experience regarding to the assessment and/or management of pain in children during the interview. From theme one, there are two sub-themes those are inadequate knowledge of pain assessment methods and inadequate knowledge of pain management practice were barriers to proper pain management in children.

*Sub-theme 1*: *Adequate knowledge of pain assessment methods*. Most of the participants describe that, their most common pain assessment method was by observing the child's facial expression or behavior like crying as well as other ways of expressing their pain level, using vital signs to verify pain, and family reports were used to determine the level of a child's pain. Similarly, most of the participants reported that we are not using the assessment tool, they overlook pain assessment tools some participants do not know there is a pain assessment tool specifically for children. One of the study participants describes as follows: -

"......At first, we will do vital signs; their temperature is included, it is a way to assess them because they are sick. Another one is that when you see them, their faces are contorted. We can understand the level of pain by looking at children, who are very sick. Especially, if children with fractures because they scream a lot. It can be understood because they do not touch their bodies. Even though children cannot say that this is what hurts them; we can understand their behavior. Otherwise, nothing is set at the manual level". (P5, age 30)

Similarly, another participant expressed "As soon as they arrive, they will be taken for vital signs and then we will take a pain assessment for example, they may have a temperature; They can make themselves unconscious. Lethargic; there may be trauma. We look at these things and give ourselves a score of pain. If the temperature is above 38.5; if so; from what mothers tell us, sometimes they cry a lot, they say that they throw up and we set a level from here. If the other is difficult and you cannot describe it, we will see the level of pain. We don't use anything". (P6, age 30)

Most of the participants reported that assessing a child's pain is difficult because they cannot express their feelings like adults so they need special skills and knowledge to assess the child's pain but now this is a big challenge. 36 years old, male participant expressed as follows;

**Table 9. Theme generation to explore the barriers affecting pediatric pain management practice for hospitalized children in Bahir Dar city public hospitals, Amhara region, Ethiopia, 2022.**

| Codes | Sub-themes | Themes |
|---|---|---|
| • Posted on the wall<br>• Using vital signs<br>• using facial expression<br>• Ways of expressing their pain level<br>• Not using an assessment tool | Adequate knowledge of pain assessment methods | Inadequate knowledge of pain assessment and pain management practice |
| • Being a child<br>• Not treated properly<br>• Not using nonpharmacological methods<br>• Complication of untreated pain | Inadequate knowledge of pain management practice | |
| • Unsatisfactory pain management practice<br>• Unsatisfactory pain grading/scaling | Satisfaction of nurses | Inadequate professional commitment and satisfaction |
| • Professional denied for decision<br>• Lack of professional interest<br>• Physicians do not accept the standard<br>• Professional efforts | Inadequate professional commitment | |
| • Availability of anti-pains<br>• Lack of assessment tools<br>• Lack of updated protocols<br>• Shortage of resources<br>• Lack of favorable infrastructure | Insufficient availability of resources in the institutions | Organization related factors |
| • Lack of training<br>• Work overload<br>• Limited experts<br>• Low monitoring and evaluation system<br>• Initiatives are focused only on seasonal | Insufficient in-service training, monitoring, evaluation, and feedback | |
| • Family cultural impact on effective treatment<br>• Family knowledge impact on effective treatment | Knowledge and culture of the family | Impacts of family knowledge, culture, and economical status |
| • Economic status of the family | Economic status of the family | |

*"According to our class, they are treated from 1 month to 14 years old, so it is difficult to distinguish the pain of those who are less than 5 years old. There is a pain that you can see, that is, facial expression and you can identify it. On top of that, they are irritable, they cry, and there is something they show you when you touch them. They become disturbed mood. They feel discomfort and most babies express their pain by crying. The family will tell you how they feel, and you can replace the ones I mentioned above, we will give you pain relief from the smallest to the most severe"*. (P8, age 36)

On the other way in this study few nurses were described, using a tool for assessing pediatric pain but it is not routine. A 39-year-old female respondent stated, *"We have an assessment tool. As for the children, we have different sections for children and adults. In pediatrics, we classify pain as no pain, mild, moderate, and severe; we will group the moderate and severe special ones separately. The reason is that we treat children with moderate pain with a high dose. This has its level of pain distribution. Painless 0, mild 1–3, moderate and high 4–10, these are aggregated to give their ranking"*. (P2, Age 39)

*Sub-theme 2*: *Inadequate knowledge of pain management practice.* Several nurses described that pain management philosophy is starting from weak opoids. As they stated, the rationale for this approach was to avoid complications such as over-sedation or respiratory depression. In addition, participants reported that children are not similar to adults so being a child is a major barrier for effective management because they do not recognize and report their pain.

Similarly, many nurses described that we use cold compress unless we are not using other non-pharmacological pain management to relieve pain. Many of the in-depth interviewees mentioned that there was managed pain simply by observing the severity of their pain but the level of pain score was not measured by specific tools appropriately due to this, the patient was not treated properly according to the degree of pain.

*"......There used to be a sedation called "PRN" now it's gone. Currently, we are giving it on a fixed schedule. There is a difference between the medicines taken and the pain score. It is not right at the level you give it, but below it and above it. Therefore, it is difficult for me to say that the right relief is given for the right pain. It is a bit difficult for me to say that the correct treatment is being given from here".* (P8, age 36)

In the other way, some nurses said that the treatment depends on the level of pain that they show the symptoms. *"Their treatment is based on the degree of severity of their pain. For example, the manifestation of pain is fever; and crying, if it is a breathing problem, we should see and measure the breathing. Children with chest pain have a sense of grunting. Therefore, if the pain is not too high, we will treat it with "Paracetamol" PRN. If the pain is high, we will prescribe BID or TID. Therefore, we will see the pain condition of the children and prescribe "PRN" BID, or TID depending on their pain".* (P7, age 34)

The non-pharmacological treatment is one of the psychological supports to relieve pain by creating a favorable environment. The majority of in-depth interviewees mentioned that they used cold compress only because the room was not comfortable to apply other non-pharmacological pain management.

*"If they have a fever, we will do cold compress. If elevation is done on trauma, we will do it. We do not do anything other than these. The rooms are not comfortable, they are overcrowded and this makes it difficult for us to provide the services we need."* (P6, age 30)

Similarly, other participant stated that *"... if they have a fever, we often use a cold compress. We do not use anything other than those mentioned, but at the family level, they play, keep them calm and do other things"* (P7, age 34)

*"In our department, there is no effort to draw like puppets and drawings. The room has problems since its layout, so it is not done, but the professional does apply a cold compress in most cases".* (P8, age 36)

Only a few participant nurses who worked at the pediatric ward stated that we use the non-pharmacological method in the inpatient room. One of the ways to make children forget their pain is to be able to attract them to make them happy. *"Non-pharmacological as we are doing the treatment in the inpatient treatment room, we have to have toys inside, so when there are these toys in the treatment. First, it thought that the children would be mentally active. Second, when there is a bicycle cart instead of medicine, at least they forget and become active, so they recover from their pain/suffering".* (P1, age 38)

**Theme 2: Inadequate professional commitment and satisfaction.** This theme contains two sub-themes: -

*Sub-Theme 1*: *Satisfaction of nurses on pain management practice.* In this study, the majority of nurses revealed that they were not satisfied with pain management practice and pain assessment scaling or pain grading. This is due to there are different factors in applying effective pain management practice.

*"I am not satisfied, because the level of pain should be set correctly and managed with the right material. They should be managing according to the signs and levels they show. There must be a tool. These things must be there, even if we give treatment with the standard screen, these things must be filled because they are basic"* (P2, age 39).

The other participants reported, **"***When we look at it as a whole, there is no satisfaction. However, I do my best, so no one can say that I did not do it. It is a little sad to see your hitters suffer due to the lack of resources and the uncomforting of the room. As a professional, I am doing my best personally" (*P5, age 30*).*

*Sub-theme 2*: Inadequate professional commitment. Some participants said that lack of professional interest; professionals delaying decisions and some physicians not accepting the assessment scale or grading affect proper management. Similarly, few nurses stated that nurses attributed the delay in nursing intervention, improper caregiving, not giving timely pain relief drugs, and delay in pain diagnosis and control due to the late presence of nurses and doctors at the patient's bedside. In addition to giving analgesics to patients, doctors must write an order sheet this affects a treatment because we cannot give immediately to recognize the pain.

*"... some doctors also seem to be delayed in making a decision. As it is a teaching hospital, it is decided by one another that they may remain in pain at this time" (*P5, age 30). Similarly, one nurse reported that *"Doctors should also know because they order /PRN/ and miss the standard, so that there is a uniform procedure, the children are the responsibility of tomorrow's country" (*P2, age 39*).*

*"The biggest thing is lack of interest. In addition, because they are not like adults, they do not have anything to tell you when they are sick or they do not want to tell you everything, so their age is the biggest thing. You must have professional motivation..."* (P8, age 36)

*"Some doctors are not accepting our "pain score"; there are situations where they don't accept what we tell them..."* (P6, age 30)

On the other side, some nurses reported that nurses should always be ready and give close follow-ups to patients.

*"It is necessary to closely monitor the condition of the children. There is no need for a nurse to be careless, therefore; I say that according to what we have learned, it should be done properly and any activities we have done should be recorded in the medicine that we have given".* (P7, age 34)

**Theme 3: Organizational-related factors.** The majority of nurses reported that insufficient availability of different resources in the institutions and insufficient in-service training, monitoring, evaluation as well as feedback were major barriers to effective pain management for hospitalized children.

*Sub-Theme 1*: Insufficient availability of resources. All participants revealed that the availability of anti-pain in the hospital, lack of updated protocols about pediatric pain management, availability of specific tools in each room, un designed (narrowing rooms) infrastructure, and shortage of materials were barriers to pain management practice.

*".... There are materials that can be filled for nursing activities, starting with thermometers, for each profession. The third is that there are many beds in each room. We are talking about illness. Class five, six, and seven beds are reserved, which destroys what we previously called mental alertness. In addition, they will bring congestion. It creates workload for nurses."* (P1, age 38)

The other participant said, *"As in the emergency department, the medicines that we call a challenge to manage pain are not always available. For example, if there is a child with very severe pain, sometimes morphine is not available in an emergency pharmacy. They can be purchased outside the hospital. Our department is not a true emergency room. We use a room with one small examination bed. When another patient arrives, it is difficult to manage because we put it on the outside seat. Even when it is prepared from the beginning, it is difficult because it is not prepared for emergency". (*P5, age 30)

*". . .The second obstacle is material shortages; it makes it difficult for us to provide quality service".* (P1, age 38) Similarly, other participant reported, *". . . absence of tools, there is no updated protocol; science is new every time. . ."* (P6, age 30)

*Sub-theme 2*: *Insufficient in-service training, monitoring, evaluation, and feedback.* The majority of study participants reported that lack of training, limited pediatric experts, work overload, weak monitoring and evaluation system of nurses, and lack of continuous follow-up for implementation of initiatives were major barriers to proper pain management practice.

*". . .. independent authority is not given to the profession. Orientation is not given to every professional according to the standard. There is no training given, but we are putting information level. There is a focal person like the unit coordinator, and that focal person also has to monitor things, especially if he gives training".* (P2, age 39)

*". . . . . . . .things to pay attention to better than before, but it is only happening seasonally, we are in trouble, but I say it is good if it has sustainability, but I say it is not just for the competition".* (P7, age 34)

*"Therefore, it is difficult for me to say that the right relief is being given for the right pain. Jobs overlap. We have limited expertise. It is a bit difficult for me to say that the correct treatment is being given to her".* (P8, age 36)

**Theme 4: Impacts of family knowledge, culture, and economical status.** In this study, more than half of the participants reported that knowledge, cultural and economic status of the family were a barrier to proper pain management.

*Sub-theme 1*: *Knowledge and culture of the family.* More than half of nurses explained that family beliefs and misconceptions are factors that influence the management of pain so our society's knowledge, thinking, and culture influence us to give appropriate treatment. Some participants explained that especially if the child is febrile, they said it is "evil eye/በ·ዓ/" Due to this they wanted to go to traditional medicine houses.

One nurse said *"Coincidentally, it was months ago and the child had a high fever. I was about to give IM anti-pain. Family will not give it, they said we would not inject it because it would be "evil eye/በ·ዓ/", and then explain the problem to them if they agree. Just as I was injecting him, there was a delay in explaining them, so he fainted and started shaking, which is what we call a febrile seizure. From that family, they directly contacted with an injection and an evil eye, and while we were talking, they said that our son had killed him. There was a lot of controversy and they thought of something else. . ."* (P7, age 34)

*"Most of the time, by connecting it to the cultural practice, there are those who say that the needle should be left. Some say let us show it from another first. At most, the evil eye may have eaten it; it is a homecoming problem, his grandfather's curse "AZINABET" explains such and*

*such things. So, in my view household knowledge level, Attitude, culture, and way of thinking prevent you from doing the right thing"*. (P8, age 36)

*Sub-theme 2*: *Economical status of the family*. The majority of nurses reported that the economic status of the family affects our treatment. This is due to some anti-pain drugs like morphine and others are not always avail inside the hospital at that time they cannot afford to buy drugs outside the hospital.

"*Economic status of patients for example, if you prescribe morphine there are people who cannot afford it, and as a result, there are situations where sedation is not given properly*" (P8, age 36).

## Discussion

Nurses have a crucial role in the assessment, management, and alleviation of patients' pain. In this study 216(53.6%) (CI- 48.4, 58.3) nurses had good practice regarding the pain management of children. This result is in line with the study in Turkey (54.5%) of study participants used a combination of pharmacological and non-pharmacological methods to relieve pain in children, in Ghana (57.8%) nurses had good pediatric pain management practices, in black lion hospitals (52.7%) and also the result observed in Mekelle city public hospitals, Ethiopia (55,8%) [9,13–15]. The possible reasons for this discrepancy might be due to differences in knowledge and differences in familiarity with pain management tools and protocols because Ghana nurses have relatively good knowledge about pediatric pain management. In addition, they were familiar with pain assessment and management tools as compared to our country.

There is evidence reported that nurses having good knowledge had good practice regarding pain management for hospitalized children compared to their counterparts [14]. In addition, the current study included almost all wards from each hospital but the study in black lion hospital was conducted only from the pediatrics unit [9]. The finding of this study was higher than the study conducted in Jimma Hospital in that the overall level of nurses' pain management practice was (36.6%) [10] and study in Amhara region referral hospital showed that (45.7%) had good practice [8]. This discrepancy might be due to nurse's commitment to applying the theoretical knowledge into actual practice due to a new initiative launched by Federal Ministry Of Health (FMOH) in recent times to create pain-free hospital initiatives so the nurses considered pain as the fifth vital sign to focus on pain management practice compared to previous study. This study is also relatively lower than the result conducted in Bahir Dar city HCPS showed that (60.6%) of HCPs had good practice [7]. This discrepancy might be due to the study participants' variation, the previous study participants were all healthcare professionals including physicians, but in this study, the study participants were only nurses. Other possible reasons for the contrary might be due to the difference in the size of the study samples and the number of hospitals included in the study.

In this study, nurses who qualified in BSC pediatrics and child health nursing had a higher practice level on pain management for hospitalized children than those nurses who qualified in the diploma level. This finding was similar to a study in an Australian pediatric hospital showed that nurses with specialist pediatric qualifications had significantly better practice scores than other nurses. Similarly, our finding was comparable with the study conducted in Hawassa university referral hospital, where nurses who had a degree and above practiced 1.9 times more pain assessment and management than diploma nurses [16,17]. This might be due to pediatrics professionals having better knowledge and practice because they learn only pediatrics health and treat only children than other professionals.

This study is also supported by qualitative findings of this study (theme 3, participant 8), some participants reported that there are limited experts qualified in child health due to that it's difficult to say that proper treatment is given by other professions.

On the contrary, a study done in Ghana revealed that critical care nurses were about 5.87 times more likely to engage in good practices compared to pediatric nurses regarding pain management whereas bachelor's degree and diploma holders were 10 and 8 times, respectively, more likely to engage in good practices compared to master's degree holders [14]. This inconsistency might be due to the difference in attending standard training and education on pediatric pain management, which means that in the study conducted in Ghana participants were more trained about pain management for hospitalized children as compared to the current study which helps them to enhance practice towards pediatrics pain management. Nurses who attended standard training regarding pain management have good practice as compared to their counterparts. These differences might be due to their frequent exposure to the management of children's pain because those nurses usually worked at the pediatrics unit, they develop better knowledge and practice than others do.

Nurses who had 2–5 years and 5–10 years of experience in the pediatrics unit were 1.9 times and 5.4 times more likely able to practice pediatrics pain management compared to those nurses who had less than two years of experience in the pediatrics unit respectively. This finding is supported by a study conducted in Hawassa university referral hospital, who had nine and above years of service and were 2.35 times more likely in pain assessment and management than those service years less than nine years [17]. This is due to experienced nurses were tending to provide more accurate pain management because experiences make them more skillful at interpreting pain scores and their management. As well as more experienced professionals become acculturated into the role of a profession, perceptions of pain, and practice of pain management.

In this study, nurses who had good knowledge of pain management were nearly four times more likely able to practice pain management for hospitalized children than those who had poor knowledge of pain management. This result was in line with the study done in England, which showed that nurses who had good knowledge of pediatric pain management had effective pain management practices as compared to their counterparts, a study in Malaysia showed that nurses who had good knowledge of pain management had a significant positive relationship with pediatrics pain management practice in Amhara region, Ethiopia showed nurses who had adequate knowledge was 2.7 times more likely to practice proper pain management for hospitalized children than those who had inadequate knowledge [8,18,19]. This study finding is also supported by qualitative findings most of the participants reported that lack of knowledge due to this we are not using the assessment tool, they overlook pain assessment tools some participants do not know there is a pain assessment tool specifically in children so this affects to practice effectively. Similarly, most of the participants reported that assessing a child's pain is difficult because they cannot express their feelings like adults so it needs special skills and knowledge to assess the Child's pain but this is a big challenge. This result implies that a nurse's knowledge can affect his or her ability to provide children's pain management adequately and nurses who have adequate knowledge assertively practice pain management

In this study, nurses who had favorable attitudes toward pain management were 3 times more likely to practice pain management for hospitalized children than those who had unfavorable attitudes. This study finding was in line with the study done in Jordanian public hospitals, where nurses who had favorable attitudes towards pain management had a significant positive relationship with their pain management practices, Study in England many nurses have negative attitudes towards pain, and pain management in pediatric populations leads to delayed administration of analgesia or none at all and study in Amhara region, Ethiopia

showed nurses who had favorable attitude towards pain management were 2.3 times more likely able to practice pain management for hospitalized children than those had unfavorable attitude [8,18,19]. The possible explanation might be those nurses who were interested in their profession and perceived the child pain suffering was enhancing pain management practice.

This study showed that a lack of in-service training for nurses is another factor that negatively affects good practice of pain management for hospitalized children. This finding is consistent with study findings from New Zealand, Amhara region Ethiopia, Debre Tabor hospital Ethiopia, and another study in Bahir Dar City showed that a lack of in-service training for nurses is a factor that affects proper pain management practice for hospitalized children [7,8,11]. The possible explanation is that non-trained health professionals did not achieve their competency in providing safe, high-quality, health services to clients through improved work performance. This is because through training different types of skills/practice are taught, including psychomotor skills, clinical decision-making skills, and communication skills. Therefore, trained health professionals in quality improvement have the potential to impact positively on attitudes, knowledge, and practice of pain management. This finding was also supported by our qualitative findings (theme 3, participant 2) showed that many of the study participants reported that lack of training was a major barrier to proper pain management practice.

Qualitatively the finding of this study (theme 3, participant 8) showed that more than half of nurses explained that family beliefs and misconceptions are factors that influence the management of pain so our society's knowledge, thinking, and culture influence us to give appropriate treatment. Some participants explained that especially if the child is febrile, they say it is an "evil eye" Due to this they wanted to go to traditional medicine houses. Similarly, the majority of nurses reported that the economic status of the family affects our treatment this is due to some anti-pain drugs like morphine and others are not always avail inside the hospital at that time they cannot afford to buy drugs outside the hospital. This study was in line with studies conducted in Europe, some clinicians stated that younger children were more difficult to assess and manage [20]. In London, nurses felt that parents exaggerate their child's pain and ask for analgesic drugs before their child needs them. Children and parents do not inform nurses when they/their child are in pain as a barrier [21]. In Canada, Inadequate patient communication with the health professional, pain experiences, and language barriers are factors in pediatric pain management [22]. At Auckland University of Technology, Parents can challenge the decisions of health professionals [23].

Similarly, the qualitative finding of this study showed that inadequate professional commitment and satisfaction of nurses were factors for proper pain management. Some participants said that lack of professional interest; professionals delaying decisions and some physicians not accepting the assessment scale or grading affect proper management. In addition, a few nurses stated that nurses attributed the delay in nursing intervention, improper caregiving, not giving timely pain relief drugs, and delay in pain diagnosis and control due to the late presence of nurses and doctors at the patient's bedside. This study finding is supported by a study conducted at a UK children's hospital [24].

## Strength and limitation of the study

### Strength

This study used a mixed study design to assess the pain management practice of nurses, so applying quantitative as well as qualitative (to explore the factors not addressed by the quantitative study) might improve the quality.

## Limitation

The study did not observe the actual practices of health care professionals' assessment and management of pain in children due to the self-reporting nature of the questionnaire the finding of this study might be influenced by subject response bias this is because of time and financial constraints.

## Conclusion

Only half of the participants had good practice in pediatrics pain management. Knowledge, attitude, nurses qualified in BSC pediatrics and child health, years of experience in the pediatrics department, getting pain management training were associated with the practice of pain management. From the qualitative findings, the availability of anti-pain drugs, lack of training, lack of assessment tools, lack of updated protocols, shortage of resources, lack of continuous monitoring and evaluation, and others were the barriers to proper pain management. This study concludes that applying effective pain management practices to hospitalized children remains a challenge. Therefore, it is better to put further effort towards improving pediatric pain management practice.

## Recommendation

Both qualitative and qualitative findings confirmed that lack of training in pediatric pain management, no having of a protocol for pediatric pain management at institutions, a lack of availability of anti-pain drugs, nurse's workload, lack of availability of pain assessment tools, and no designated area and material for pain management contributes to low level of pain management practice of nurses. Based on the findings of this study the following recommendations are forwarded: -

### For ministry of health and regional health Bearues

❖ To implement a pain-free hospital initiative, should support continuous follow-up, feedback, monitoring, and evaluation rather than seasonal or computation/rank/ based.

### For hospital administrators

❖ The hospital's administrative bodies would better provide appropriate professional education, and training programs regarding pain management for staff nurses, fulfill all analgesics used to treat pain, and for each hospital to build well-ventilated and attractive playing rooms with well-equipped materials to apply non pharmacological interventions.

### For nursing leaders (metron /nursing directors)

❖ It would be much better if the metron, staff nurses, and academic nursing staff in conjunction with the regional health bureau work together on major factors and execute means to alleviate those major constraints.

### For universities and colleges

❖ Education is a central aspect of strengthening knowledge, attitudes, and practice of pediatric pain, so universities and colleges should focus on undergraduate pediatrics and child health nurses for better children's health.

### For researchers

❖ Furthermore, researchers better use observation or checklist methods of data collection techniques.

## Supporting information

**S1 File. Quantitative questionnaire.**
(PDF)

**S2 File. Qualitative questionnaire.**
(PDF)

## Acknowledgments

The authors would like to thank the hospital directors and the heads of each working unit for their kind assistance during data collection. The authors also would like to thank all of the study participants who participated in this study.

## Author Contributions

**Conceptualization:** Bekele Berihun, Zeamanuel Anteneh Yigzaw.

**Data curation:** Bekele Berihun, Lakew Asmare.

**Formal analysis:** Netsanet Fentahun.

**Investigation:** Bekele Berihun.

**Methodology:** Netsanet Fentahun, Lakew Asmare.

**Software:** Netsanet Fentahun, Lakew Asmare.

**Supervision:** Zeamanuel Anteneh Yigzaw.

**Validation:** Bekele Berihun, Lakew Asmare.

**Writing – original draft:** Bekele Berihun.

**Writing – review & editing:** Zeamanuel Anteneh Yigzaw.

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
