## [Decision Letter · Decision Letter 0]

21 Feb 2024

PONE-D-23-42900Pediatrics Pain Management Practice and Factors Associated with Hospitalized Children among Nurses Working in Bahir Dar City Public Hospitals, Northwest Ethiopia, Mixed MethodPLOS ONE

Dear Dr. yigzaw,

Thank you for submitting your manuscript to PLOS ONE. After careful consideration, we feel that it has merit but does not fully meet PLOS ONE’s publication criteria as it currently stands. Therefore, we invite you to submit a revised version of the manuscript that addresses the points raised during the review process.

We had one only reviewer with minor suggestions, which I agree and  consider enough  if you can do the revision and submit for a final decision.  Please make a revision on the writting, preferently by a native speaking person, which will also improve the manuscript. 

We look forward to receiving your revised manuscript.

Kind regards,

Ricardo Q. Gurgel, PhD

Academic Editor

PLOS ONE

2. In the ethics statement in the Methods, you have specified that verbal consent was obtained. Please provide additional details regarding how this consent was documented and witnessed, and state whether this was approved by the IRB.

Reviewers' comments:

Reviewer's Responses to Questions

**Comments to the Author**

1. Is the manuscript technically sound, and do the data support the conclusions?

Reviewer #1: Yes

2. Has the statistical analysis been performed appropriately and rigorously? 

Reviewer #1: Yes

3. Have the authors made all data underlying the findings in their manuscript fully available?

Reviewer #1: Yes

4. Is the manuscript presented in an intelligible fashion and written in standard English?

Reviewer #1: Yes

5. Review Comments to the Author

Reviewer #1: Dear Author,

This mixed method study which aimed to assess practice and factors associated with pediatric pain management among nurses working in Bahir Darcity public hospitals is well designed and performed, only some suggestions are listed below:

- Title: the title is too long and it could be most clear for readers. Suggestion: "Practice and Factors Associated with Pediatrics Pain Management among Nurses Working in Bahir Dar City Public Hospitals: a mixed method study";

- Introduction: this section is too long. It is suggested that the main topoics related to the theme developed in the study should be exposed. It is suggested that lines 69-92 from page 4 and line 98 -107 from page 5 should be removed, once there is a subjetive information not directly related to the theme;

- Materials and Methods: On this topic, there is also superficial information that must be removed. Information about the cities presented on lines 151-165 from page 7-8, is not necessary to be written in this paper. In this same way, detalis about the sample size calculator explained on page 8, is not necesssary to be in this section;

Based on the aspects analyzed and listed above, this paper should be minor revised for publication on PlosOne.

Best regards

6. PLOS authors have the option to publish the peer review history of their article (what does this mean?). If published, this will include your full peer review and any attached files.

Reviewer #1: No

---

## [Author Response · Author response to Decision Letter 0]

2 Mar 2024

Dear Reviewers

Greetings.

Thank you very much for your support

We do very important research with great public health importance. The research is the first study in the study area. We address all the points raised by the reviewers. We do the research with a legal research ethical letter (attached). If you have other review comments, we are ready to amend them again.

Plos one was ideal for this mixed-method research.

We hope we are waiting for this publication in this journal.

Thank you very much for your support and understanding.

---

## [Editor Report · Decision Letter 1]

6 Mar 2024

Practice and Factors Associated with Pediatrics Pain Management among Nurses Working in Bahir Dar City Public Hospitals: A Mixed Method Study

PONE-D-23-42900R1

Dear Dr. Yigzaw,

We’re pleased to inform you that your manuscript has been judged scientifically suitable for publication and will be formally accepted for publication once it meets all outstanding technical requirements.

Kind regards,

Ricardo Q. Gurgel, PhD

Academic Editor

PLOS ONE

Additional Editor Comments (optional): Changes were done and the manuscript has improved considerably, deserving publication.
---

## [Editor Report · Acceptance letter]

25 Apr 2024

PONE-D-23-42900R1 

PLOS ONE

Dear Dr. Yigzaw, 

I'm pleased to inform you that your manuscript has been deemed suitable for publication in PLOS ONE. Congratulations! Your manuscript is now being handed over to our production team.

Kind regards, 

on behalf of

Professor Ricardo Q. Gurgel 

Academic Editor

PLOS ONE